# Functions of Phytochrome Interacting Factors (PIFs) in Adapting Plants to Biotic and Abiotic Stresses

**DOI:** 10.3390/ijms25042198

**Published:** 2024-02-12

**Authors:** Zhao-Yang Li, Ning Ma, Fu-Jun Zhang, Lian-Zhen Li, Hao-Jian Li, Xiao-Fei Wang, Zhenlu Zhang, Chun-Xiang You

**Affiliations:** 1College of Horticulture Science and Engineering, State Key Laboratory of Wheat Improvement, Shandong Agricultural University, Tai’an 271000, China; zyli_1106@163.com (Z.-Y.L.); mning1006@163.com (N.M.); 17863805279@163.com (F.-J.Z.); llzhen0620@163.com (L.-Z.L.); 13455616801@163.com (H.-J.L.); xfwang2004@163.com (X.-F.W.); 2Department of Horticulture, College of Agriculture, Shihezi University, Shihezi 832003, China

**Keywords:** phytochrome interacting factors, biotic stress, abiotic stresses

## Abstract

Plants possess the remarkable ability to sense detrimental environmental stimuli and launch sophisticated signal cascades that culminate in tailored responses to facilitate their survival, and transcription factors (TFs) are closely involved in these processes. Phytochrome interacting factors (PIFs) are among these TFs and belong to the basic helix–loop–helix family. PIFs are initially identified and have now been well established as core regulators of phytochrome-associated pathways in response to the light signal in plants. However, a growing body of evidence has unraveled that PIFs also play a crucial role in adapting plants to various biological and environmental pressures. In this review, we summarize and highlight that PIFs function as a signal hub that integrates multiple environmental cues, including abiotic (i.e., drought, temperature, and salinity) and biotic stresses to optimize plant growth and development. PIFs not only function as transcription factors to reprogram the expression of related genes, but also interact with various factors to adapt plants to harsh environments. This review will contribute to understanding the multifaceted functions of PIFs in response to different stress conditions, which will shed light on efforts to further dissect the novel functions of PIFs, especially in adaption to detrimental environments for a better survival of plants.

## 1. Introduction

Light is a key environmental factor driving carbon metabolism, which is closely involved in almost every facet of growth and progression in plants [1,2]. Multiple light receptors have been evolved in perceiving the surrounding light signals [3]. Phytochromes (phys) are among these receptors that perceive red (R) and far-red light (FR) light signals with wavelengths ranging from 600 to 750 nm. In *Arabidopsis*, five phytochrome family members have been identified, ranging from phytochrome A–E (phyA–phyE) [4,5,6]. Phytochromes have an inactive form (Pr) for R light absorption and an activated form (Pfr) for FR light absorption [7,8]. When exposed to R light, the phytochromes in inactive Pr form are converted into activated Pfr form and translocate from the cell cytoplasm to the nuclear compartment to interact with downstream regulators [9].

PIFs are among the most important downstream factors interacting with the Pfr form of phytochromes [10]. Light-activated phyB induces rapid (within minutes) phosphorylation of PIF3, which is subsequently ubiquitinated and targeted for degradation via the 26S proteasome complex, resulting in the initiation of photomorphogenesis [11]. PIFs are typical basic helix–loop–helix (bHLH) TFs that have already been characterized in multiple plant species, including *Arabidopsis thaliana*, rice (*Oryza sativa*) [12], tomato (*Solanum lycopersicum*) [13], maize (*Zea mays*) [14], apple (*Malus × domestica*) [15], wheat (*Triticum aestivum* L. cv. Chinese Spring) [16], populus (*Populus trichocarpa*), oriental melon (*Cucumis melo* L.), and cotton (*Gossypium hirsutum*) [17]. The phylogenetic relationship of PIF orthologs with multiple plant species were analyzed and revealed that these PIFs can be classified into five groups (Groups I–V) (Figure 1A).

In plants, an active phytochrome B-binding (APB) domain and a bHLH domain are commonly present in PIFs (Figure 1B) [15,18]. However, an active phytochrome A-binding (APA) domain may also be found in certain PIFs, such as AtPIF1 and AtPIF3 in *Arabidopsis*, and several orthologs in apples, namely MdPIF1–MdPIF5 [19], as well as in other plant species including wheat (TaPIF1), corn (ZmPIF1 and ZmPIF3), tomato (SlPIF1 and SlPIF3), cotton (GhPIF1 to GhPIF3), poplar (PtPIF1 and PtPIF3), and melon (CmPIF3) (Figure 1B). Overall, the APA domain is generally conserved in PIF1 and PIF3 across various plant species (Figure 1B). An online website MEME (https://meme-suite.org/meme (accessed on 25 January 2024)) was used for the conserved motif analysis of PIF protein sequences. It was revealed that motif 1, motif 2, and motif 3 constitute the bHLH structural domain, which are conserved across all PIF proteins (Figure 1C). Motif 4, which represents the APB domain, is also conserved among all tested PIFs (Figure 1C). In addition, motif 11 represents the APA domain, but it only presents in certain PIF proteins (Figure 1C). With these critical domains, PIFs were determined to be key regulators in multiple growth and developmental stages in plants, such as seed sprouting [20], hypocotyl extension [21], stem branching development [22], shade avoidance response [23], biological clock [24], and flowering time [17] in plants.

As sessile organisms, plants continuously face various biotic (microbes and pests) and abiotic (temperature, salinity, and drought) stresses, which severely impact their survival [25,26]. In their adaption to these harsh environments, a series of strategies have been developed in higher plants. Among these, PIF-mediated regulatory pathways have drawn much attention and an increasing body of evidence has demonstrated the crucial roles of PIFs in response to diverse environmental triggers in plants [11,15,18,27]. In this discussion, we encapsulate the latest discoveries in which the critical roles of PIFs in regulating plant responses to stresses caused by abiotic factors (drought, low, and high temperature and salinity) are dissected. We also discuss the critical roles of PIFs in plant disease resistance reported in several newly published studies. We aim to emphasize the crucial roles of PIFs beyond the transition from skotomorphogenesis to photomorphogenesis, and provide a comprehensive understanding for how PIFs respond to stress in plants.

## 2. PIFs Are Involved in Regulating Drought Stress Tolerance in Plants

The adaptation of plants to drought consists of complex biological processes which are in close association with the crosstalk among multiple signaling pathways [28,29,30,31]. PIFs play a key role in regulating drought stress tolerance, while the dehydration responsive element binding (DREB) proteins [32,33,34,35] and the abscisic acid (ABA) are closely involved in these processes [36,37,38,39] (Table 1).

Treatment with drought inhibits the transcription of *OsPIF14*, whose product inhibits the rice *OsDREB1B* expression by directly binding to its promoter [40]. OsDREB1 has been previously reported to positively regulate the accumulation of various soluble sugars and free proline, and these accumulated osmoprotectants may contribute to the increased salt tolerance in transgenic rice [40,80]. These findings suggest that PIFs may regulate drought stress tolerance via DREB-mediated pathways. Moreover, there is a critical role for PIFs in alleviating drought-induced dwarf phenotype [41,81,82,83,84]. When exposed to drought stress, the cell number and size are reduced via the inhibited transition of the cell cycle from G1 to S phase in plants [85,86] and by modulating the transcription of genes involved in cell wall synthesis and development [87]. The ectopic expression of *OsDREB1A* enhances the drought tolerance but induces the dwarf phenotype in *Arabidopsis* [41]. The transgenic plant harboring both *OsDREB1A* and *OsPIL1* (*Oryza sativa* phytochrome-interacting factor-like 1) showed not only enhanced drought tolerance as that of *OsDREB1A* overexpressors, but also displayed promoted hypocotyl elongation and floral induction [41]. Moreover, OsPIL1 activates the cell wall tissue formation and fiber bundle synthesis, thereby increasing the size of cells and extending the internodes in transgenic rice [42]. In addition, drought inhibits the *OsPIL1* transcription, resulting in a shorter internode and reduction in plant height [41,42]. Thus, OsPIL1 functions as a key regulator in response to drought by modulating the internode elongation and plant height [42].

Additionally, PIFs also regulate plant drought tolerance by modulating the ABA signaling pathways. The ectopic expression of *ZmPIF1* and *ZmPIF3*, two PIF orthologs from maize (*Zea mays*), increases the drought resistance in genetically modified rice plants [43,44]. Further investigation shows that both PIF orthologs from maize participate in ABA signaling, as well as in regulating the stomatal aperture in rice, suggesting that PIFs play a key role in ABA-mediated stomatal aperture to modulate water-loss and drought tolerance [43,44,45]. Moreover, PIFs also regulate the drought stress responses in dicotyledons. For instance, DcPIF3 inhibits drought-induced reactive oxygen species (ROS) burst and enhances the expression levels of genes associated with ABA synthesis, leading to an increased content of endogenous ABA and promoting the drought stress tolerance in carrot (*Daucus carota* L.) [46]. Similarly, MfPIF1 amplifies the transcription of ABA-regulated downstream genes and promotes the drought tolerance in *Myrothamnus flabellifolia* [47]. Moreover, NbPIF1/NtPIF1 suppresses the expression of genes involved in ABA biosynthesis and signaling, as well as the genes related to carotenoid synthesis, leading to reduced drought tolerance for tobacco (*Nicotiana tabacum* L.) [48]. Therefore, PIFs are among the key regulators that help plants adapt to arid environments by regulating various signaling pathways, like ABA.

## 3. Key Roles of PIFs in Regulating Low-Temperature Responses in Plants

Low temperature severely impairs plant growth, development, and crop productivity, as well as limits the crop geographical distributions. Upon exposure to low temperatures, cytoskeleton rearrangement and membrane fluidity alteration are among the upstream cellular responses, followed by calcium influx which stimulates multiple cellular responses, including transcriptome regulation and the production of secondary metabolites [88,89]. It has been well established that the TFs C-repeat binding factors (CBFs) function as “molecular switches” in cold regulatory networks in plants [90,91].

The CBF-dependent regulatory pathways are crucial for cold tolerance in plants [92,93,94], and PIFs have been shown to play a significant role in these proceedings. For example, SlPIF4 (*Solanum lycopersicum* phytochrome interacting factor 4) is accumulated under low temperature, and it could increase the cold tolerance by activating the *SlCBF1* expression via interacting with the G-box motif within its regulatory region [49]. Similarly, OsPIL16 increases the transcription of the CBF family gene *OsDREB1* to promote cold adaption by inhibiting the lipid peroxidation and malondialdehyde (MDA) accumulation [50]. In contrast, other PIF orthologs, including PIF1, PIF3, PIF4, PIF5, and PIF7, could repress the transcription of *CBF*s by associating with the G-box or E-box motifs located in their promoter regions, leading to compromised cold stress tolerance mediated by the photoperiod in *Arabidopsis* [51,52,53]. Additionally, CBFs (CBF1, CBF2, and CBF3) physically interact with PIF3, leading to increased protein stability for both PIF3 and phyB [52]. Increased phyB accumulation subsequently promotes the degradation of the cold-stress negative regulators (PIF1, PIF4, and PIF5) and enhances the transcription of multiple cold-regulated genes *bzr1–1D suppressor 1* (*BZS1*), *repressor of GAI-3 mutant Like 3* (*RGL3*), and *zinc finger of Arabidopsis thaliana* (*ZAT10*), resulting in enhanced cold hardiness in *Arabidopsis* [52]. Therefore, in the CBF-dependent pathways, PIFs could interact with CBFs at both protein and transcription levels to regulate tolerance to low temperature in plants.

In addition to the CBF-dependent pathways, PIFs also regulate the cold response signaling and how plants adapt to cold via CBF-independent pathways [95,96] (Table 1). For instance, SlPIF4 could associate to the G-box in the promoter of *Gibberellic Acid Insensitive 4* (*SlGAI4*), which encodes a DELLA protein that promotes tolerance to low temperature in tomato (*Solanum lycopersicum* L) [49]. Moreover, SlPIF4 also promotes jasmonic acid (JA) and ABA synthesis but represses gibberellin (GA) biosynthesis under low temperature, suggesting that these phytohormones might be associated, at least partially, with SlPIF4-medited cold tolerance [49]. In contrast, a recent report revealed that SlPIF4 reduced the low-temperature tolerance of tomato anthers by regulating the tapetum development [54]. Specifically, dysfunctional tapetum 1 (SlDYT1) is a direct upstream regulator of defective in tapetal development and function 1 (SlTDF1), while both of them are closely involved in programmed cell death (PCD) and the development of tapetum [54,97]. Furthermore, SlPIF4 forms a complex with SlDYT1 via physical interactions to activate the transcription of *SlTDF1* at mildly low temperatures, leading to pollen abortion [54]. These observations imply that PIFs might regulate the tolerance to a low temperature through different pathways in different plant organs, resulting in different and even opposite outputs in plant cold adaption.

PIFs may also regulate tolerance to low temperature through other pathways. For example, CsPIF8 directly activates the expression of superoxide dismutase (CsSOD)-encoding gene, leading to enhanced clearance of superoxide anions and promoted cold tolerance in *Citrus sinensis* [55]. MdPIF3 negatively regulates cold tolerance by increasing ROS levels and electrolyte leakage in apples (*Malus domestica*) [56]. Therefore, PIFs are among the key regulators of plants’ adaptation to low temperature, and their functions and mechanisms may vary among different plant species and organs.

## 4. Role of PIFs in High-Temperature Stress

Global climate change has become a significant concern as it exacerbates the detrimental effects of high temperatures, resulting in a reduction in crop yield and quality [98,99,100]. As critical plant TFs, PIFs have also been determined to be key regulators in modulating plant responses to high temperature, especially for PIF4. For instance, PIF4 interacts with the promoter and stimulates the transcription of *heat shock factor A2* (*HSFA2*), a fundamental controller of heat stress adaptation, resulting in improved resistance to heat stress [57,101,102]. Importantly, PIF4 could integrate various signals such as light, biological clock, and hormonal signaling pathways to modulate plant thermomorphogenesis [57,103,104].

Recent reports have revealed that temperature perception or thermomorphogenesis is tightly linked with light perception mechanisms, and PIF4 acts a key node mediating their crosstalk. For example, a high temperature (28 °C) reduces the activity of activated form of phyB (Pfr) and results in the increased protein accumulation of PIF4, which subsequently enhances the transcription of *Oresara1* (*ORE1*) together with ethylene and ABA signaling, leading to high-temperature-induced leaf senescence [58]. Moreover, a set of suppressor of PhyA-105 (SPAs) plays a key role in stabilizing PIF4 under a high ambient temperature [105,106]. It has determined that SPAs stabilize the PIF4 in two pathways: SPAs promote the degradation of phyB, which is involved in disrupting PIF4, and SPA1 could directly interact with and phosphorylate PIF4 to enhance its stability [105,106]. Additionally, cryptochromes (CRYs) are receptors of blue light, and PIF4 also participates in blue light-regulated thermomorphogenesis [107]. Specifically, blue light promotes the interaction between CRY1 and PIF4, leading to a reduction in the transactivation activity of PIF4 [108]. Moreover, an E3 ligase constitutively photomorphogenic 1 (COP1) could interact with long hypocotyl in far-red 1 (HFR1) to promote its degradation [103,105]. While HFR1 could interact with PIF4 to inhibit its transcription activity [107,109], CRY1 could indirectly suppress PIF4′s transcriptional regulatory activity by increasing the accumulation of HFR1 via inhibiting the E3 ligase activity of COP1 [107]. Therefore, PIF4 serves as a convergence point that combines light and temperature inputs to control the growth and development of plants.

Under high-temperature conditions, plants display phenotypical changes like premature aging [58], leaf drooping [110], hypocotyl elongation [111,112], and early flowering [113], and PIF-regulated hormonal signaling are closely associated with these processes. For example, in high-temperature-induced leaf hyponasty and hypocotyl elongation, however, flowering phenotypes that are not early are abolished in the *Arabidopsis* mutant lacking *PIF4*, suggesting its pivotal function in the compensatory reactions induced by elevated temperature [59]. Moreover, PIF4 and PIF5 efficiently accelerate high-temperature-induced leaf senescence by directly modulating the transcription of their targets, including *Nam, Ataf and Cuc 019* (*NAC019*), *indole-3-acetic acid inducible 29* (*IAA29*), *CBF2*, and *senescence-associated gene 113* (*SAG113*), as well as by integrating multiple hormone signaling pathways [60]. In addition, PIF4 functions as a TF to stimulate the transcription of multiple genes, including *cytochrome P450 79B2* (*CYP79B2*), *flavin-containing monooxygenase 8* (*YUC8*), and *Trp aminotransferase of Arabidopsis 1* (*TAA1*), that are related to auxin production to enhance the hypocotyl elongation triggered by elevated temperature [61,62].

Additionally, PIF4 is also involved in restricting the stomatal production and leaf size regulated by elevated temperature [63]. Specifically, SPEECHLESS (SPCH) is a bHLH TF that is involved in stomatal lineage initiation. Upon exposure to elevated temperature, PIF4 accumulates in the stomatal precursor cells and subsequently represses the transcription of *SPCH* to regulate stomatal production [63]. In addition, PIF4 and Teosinte branched1/Cycloidea/Pcf4 (TCP4) function together to induce the inhibition of the leaf size at high temperatures [64,65]. Specifically, PIF4 directly interacts with the promoter of the cell cycle inhibitor kip-related protein 1 (KRP1) and enhances the transcription of its encoding gene, leading to suppressed cell proliferation and leaf size [65]. However, the binding of PIF4 to the *KRP1* promoter requires the presence of TCP4, which is not only an interacting partner of PIF4 but also a positive regulator for the transcription of *PIF4* [64,65].

Therefore, PIF4 has been determined to be a fundamental component of high-temperature signaling [59] and its roles in adapting plants to high-temperature stress have been extensively investigated. However, whether other PIF orthologs possess similar functions requires more efforts to elucidate.

## 5. Role of PIFs in Salt Stress

Soil salinization severely restricts crop harvest and standard, which has become an increasing threat to agriculture across the world [114]. Salinity exerts osmotic and toxic pressures on plants that result in growth inhibition, developmental alterations, metabolic adjustments, ion sequestration or exclusion, and finally leading to compromised crop yield and quality [115,116]. In adaption to the salinity condition, plants have developed a series of strategies, including activating osmotic stress pathways, the regulation of ion homeostasis, involvement of hormonal signaling pathways, regulation of cytoskeleton dynamics and the cell wall composition [116] (Table 1).

PIFs have been determined to regulate responses to salinity through different mechanisms. For example, the salt tolerance of the quadruple mutant *pif1/3/4/5* is promoted compared to that in wild-type, suggesting that PIFs are redundant negative regulators of salt tolerance in *Arabidopsis* [117]. Further investigation shows that PIF4 directly interacts with the promoter and represses the transcription of the NAC family gene *Jungbrunnen 1* (*JUB1/ANAC042*), and thereby indirectly inhibiting the transcription of the downstream salt tolerance gene *DREB2A* to negatively regulate the salt tolerance [66]. Additionally, PIF4 can directly bind to the salt stress-responsive negative regulatory genes *Oresar* (*ORE1/ANAC092*) and *senescence-associated gene 29* (*SAG29*), promoting their expression and exacerbating the susceptibility to salinity in *Arabidopsis* [66,67,68].

The conserved salt-activated salt overly sensitive (SOS) pathway is a classical and crucial pathway in adapting plants to salt stress [118,119,120]. Briefly, salt-induced calcium signals are decoded by SOS3, which binds calcium ions and subsequently activates and recruits the protein kinase SOS2 to the cell membrane. Then, SOS2 catalyzes the phosphorylation and subsequent activation of SOS1, which is a Na^+^/H^+^ antipoter that transports excess Na^+^ out of the cell [118]. Recently, it was reported that light promoted the salt tolerance of *Arabidopsis* seedlings compared to that in dark. Specifically, light-activated phyA and phyB interacted with and enhanced the salt-activated kinase activity of SOS2, which subsequently interacted with and phosphorylated PIF1/3 in the nucleus, leading to the decreased accumulation of PIF1/3 and enhanced salt tolerance of *Arabidopsis* seedlings [70]. Interestingly, the same group also reported that SOS2 phosphorylated PIF4/5 and suppressed the interaction between phyB and PIF4/5, resulting enhanced accumulation of PIF4/5, leading to the increased sensitivity to low light conditions and elongated growth in *Arabidopsis* [71]. These data indicate the critical role of SOS2-PIFs in different cellular processes.

In contrast to the negative roles in *Arabidopsis*, PIFs may also function as positive factors that enhance the tolerance to high salinity in diverse plant species [44,72]. For example, MfPIF1/8 enhances salt resistance by boosting the activity of antioxidative enzymes, maintaining lower levels of ROS and MDA in *Myrothamnus flabellifolia* [47,73]. Similarly, CaPIF8 positively modulates the tolerance of chili pepper to salt, because repressing the *CsPIF8* expression results in severe damage to chili pepper plants under high salt concentration in the soil, and the increased ion leakage and decreased ABA content might contribute to this [72]. Moreover, OsPIL14 directly interacts with the promoter of *Expansin 4* (*OsEXPA4*), a cell elongation-related gene, to promote its transcription, thus, the overexpressing *OsPIL14* improves the growth of mesocotyl and the root of rice upon exposure to salinity [69]. Paradoxically, treatment with salt facilitates the transcription of *OsPIL14* but promotes the degradation of OsPIL14 through 26S proteasomes, probably facilitating the turnover of OsPIL14 in rice [69]. Additionally, salt stress promotes the accumulation of Slender Rice 1 (OsSLR1), a DELLA protein that negatively regulates salt tolerance, which interacts with OsPIL14 and interferes the transactivation activity of OsPIL14 on genes implicated in the regulation of cell size and elongation [69]. Therefore, OsPIF14 functions together with OsSLR1 to modulate the seedling growth in adaption to salt, which improves our knowledge of crop adaptation to salt stress [69].

## 6. Role of PIFs in Biotic Stress

A plethora of environmental factors are detrimental to the survival and reproduction of plants, and herbivores and microbial pathogens are among these environmental cues [121]. To counter act these abiotic stresses, plants adopt a series of defensive pathways and activate a suite of molecular and cellular processes, including the upstream mitogen-activated protein kinases (MAPK) signaling cascades, and the downstream hormonal signaling pathways and transcriptional reprogramming [122]. Initially identified as a core component in light–phytochrome signaling pathway, PIFs are also determined to be closely associated with plant disease resistance.

JA, ethylene (Eth), and salicylic acid (SA) are crucial phytohormones related to plant disease resistance, and PIFs have been revealed to regulate plant disease resistance by participating in these pathways. For example, PIFs (PIF1, PIF3, PIF4, and PIF5) negatively regulate disease resistance to *Botrytis cinerea* by repressing the transcription of multiple defensive genes involved in JA/Eth signaling pathways, including *plant defensin1.2* (*PDF1.2*), *octadecanoid-responsive Arabidopsis 59* (*ORA59*), and *ethylene responsive factor 1* (*ERF1*) [74]. Moreover, both ethylene insensitive 2 (EIN2) and coronatine insensitive 1 (COI1) are associated with PIFs (PIF1, PIF3, PIF4, PIF5)-regulated defense responses against *B. cinerea*, indicating that PIFs function together with JA/Eth signaling pathways to control defense responses in plants [74,75,76]. The *sulfotransferase* (*ST2a*) gene encodes a sulfotransferase, which suppresses the JA signaling by catalyzing active OH-JA into inactive JA sulfate (HSO4-JA) [123]. PIF4 directly activates the transcription of *ST2a* by interacting with its promoter, and therefore negatively regulates the JA-mediated defense response [77,124,125]. In addition, PIFs also participate in SA signaling pathways to regulate plant immunity. For instance, light-inducible transcription factor CmWRKY42 directly activates the transcription of *isochorismate synthase* (*ICS*), a key biosynthetic gene of SA, leading to accumulated SA and promoted disease resistance to powdery mildew in oriental melon (*cucumis melo* var. *makuwa* Makino) [78]. However, CmPIF8 could interact with the promoter of both *CmWRKY42* and *CmICS* to repress their expression, resulting in a decreased SA content and compromised resistance to powdery mildew [78].

MAPK signaling cascades are widely involved in various life processes of plants [126], with a close relationship to plant immune response in particular, and PIFs have been shown to regulate plant disease resistance as substrates of MAPKs. In *Arabidopsis*, PIF3 represses the expression of defense genes, such as *iron-deficiency overly sensitive 1* (*IOS1*) and *Jasmonate ZIM-domain* (*JAZ*), and inhibits disease resistance to *Pseudomonas syringae* DC3000 [127]. Further investigation shows that MPK3/6 interacts with and phosphorylates PIF3, leading to enhanced repression defense-related genes and increased susceptibility to the bacteria [127].

The shade and the invasion of insects or microbes usually jointly affect plant survival, therefore, plants need to balance the interactions between shade avoidance and defense responses to achieve a better survival [128,129]. PIFs have been revealed to be involved in this interaction by modulating resource allocation to coordinate plant defense and development [130,131]. It is reported that when shaded plants are challenged with pathogens or herbivores, growth is prioritized over defense responses [130,131,132,133]. Specifically, under the concurrence of pest infestation and shading, PIF4 suppresses the JA signaling pathway by repressing the transcription of *Myelocytomatosis* (*MYC*), the master regulator in the pathway [130]. Meanwhile, PIF4 actively enhances the expression of *YUC* and *phytochrome kinase substrate* (*PSK*), two IAA biosynthetic genes, thereby promoting plant growth in a manner that evades shading [130]. In addition, the growth-inhibiting factor DELLA functions to disturb the role of PIFs in promoting growth [134]. However, the shading-induced inhibitory effect on the activity of phyB leads to a decreased accumulation of DELLA protein, which subsequently releases PIFs for exerting their functions in promoting plant growth [135]. Therefore, shading promotes PIFs activity in two possible pathways: shading inhibits phyB activity, which inhibits the phyB-mediated degradation of PIFs; and shading-inhibited phyB reduces the accumulation of DELLA proteins, which relieves the repression of DELLAs on PIFs. [132,135,136,137,138].

A suitable temperature promotes plant growth and accelerates developmental transitions, which requires a reduction in the intensity of defense response [139,140]. The balance between growth and immunity is compromised under high temperature, and PIF4 has been shown to be a core component that coordinates heat-induced growth and immunity. Specifically, mutating suppressor Of NPR1-1 (SNC1), a nucleotide-binding and leucine-rich repeat (NB-LRR) protein, triggers persistent immune reactions leading to apparent developmental abnormalities [141]. However, elevated temperature represses the constitutive transcription of *PRs* (*PR1* and *PR5*) in *snc1-1* mutants in a PIF4-dependent manner, suggesting PIF4 sits at the crossroad mediating growth and immunity under high temperature conditions [139,142]. In addition, PIF has also been found to play a role in antiviral defense through RNA interference (RNAi). RNA silencing or RNAi is a conserved antiviral mechanism found in all eukaryotic organisms. In *Nicotiana benthamiana*, the near-infrared light inducible TF NbPIF4 activates the transcription of *RNA-DEPENDENT RNA POLYMERASE 6* (*RDR6*) and *ARGONAUTE 1* (*AGO1*), thereby initiating the downstream RNAi antiviral pathway to defend against viral infection [79].

## 7. Summary and Prospects

With the growing population and rising living standards, the demand for food and crop productivity are increasing as well [143]. Currently, in the cultivation of crops, a substantial amount of resources is consumed to deal with climate change, abiotic stresses, and biotic stresses [144,145,146,147]. Stress-responsive genes have been successfully applied to enhance tolerance to corresponding stress, but they usually induce various negative effects on plant development [148]. Therefore, new alternatives that may function to balance the growth and defense are needed, and multiple investigations have proved the key roles of PIFs in balancing both crop productivity and stress tolerance [18,27,149]. Here, we describe the recent advances that demonstrate the functions of PIFs in adapting plants to abiotic stresses (drought, low, and high temperature, and salinity) and biotic stresses, besides the well-investigated photoregulated pathway (Figure 2 and Figure 3). This evidence suggests that PIFs may be a signal hub to integrate various stimuli to reprogram the transcriptional network that manipulates both stress responses and growth of plants. This is particularly the case for PIF4, which is a crucial regulator responding to biotic stresses, drought, high temperature, and salinity, especially for the latter two conditions [104,142,150]. Moreover, PIFs may function differently, even in opposition, across various plant species. For example, PIFs decrease tolerance to salt in *Arabidopsis* [66,67,68,117], but promote the salt tolerance in other plant species, such as *Myrothamnus flabellifolia* [47], chili pepper [72], and rice [69].

However, despite the fact that the functions and mechanisms of PIFs have been determined, especially for PIF4 [61,125,150,151], many of them are investigated in *Arabidopsis*. Whether PIFs function similarly in other plants, especially in agricultural crops, and the accurate mechanisms remain poorly understood. However, it is predictable that, as bHLH domain-containing TFs, PIFs will undoubtedly regulate the transcription of targets to modulate stress responses in plants [40,42,48,49,50]. With the development of new technologies, such as DNA affinity purification sequencing (DAP-seq), new targets of PIFs will be discovered and the functions of PIFs will be expanded in the near future. In addition, the function of PIFs can also be modulated by upstream factors, which may affect their protein stability or transcriptional regulatory ability [134,152,153]. For example, HFR1 could affect the transcriptional activity of PIF1 [109], while protein kinase MPK3/6 and SOS2 could phosphorylate PIF3 and PIF1/3 to regulate their protein stability, respectively [127]. Therefore, the identification of PIF-interacting partners may help discover new pathways associated with the function of PIFs in multiple plants, especially for agricultural crops.

Given the crucial functions of PIFs in balancing plant stress responses and growth, novel members, as well as novel functions, of PIFs require further efforts to explore. Recent advancements in technologies including next-generation sequencing [154], multi-omics analysis [155], clustered regularly interspaced short palindromic repeat (CRISPR)/Cas9 gene editing [156], and genome-wide association studies (GWAS) [157] have provided researchers with powerful tools to extensively explore the functions of the PIF family. It is reasonable to predict that future studies will undoubtedly enhance our understanding of the PIF family and its potential applications in agriculture and plant breeding, particularly in the pursuit of increasing yields, enhancing quality, optimizing agricultural resource utilization, and boosting tolerance to biotic and abiotic stressors [145].

## Figures and Tables

**Figure 1 ijms-25-02198-f001:**
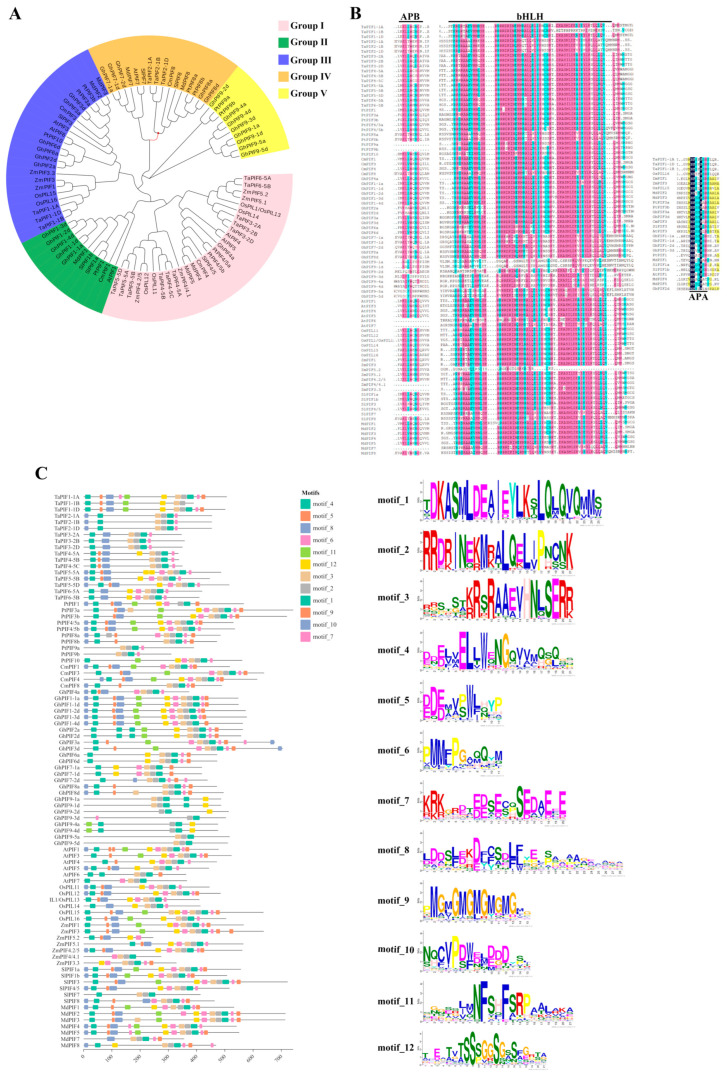
Bioinformatics analysis of PIFs proteins. (**A**) The phylogenetic evolutionary tree of PIF proteins, constructed using MEGA-X, was divided into five groups (Groups I–V), and was further esthetically enhanced using Evolview. (**B**) The PIF proteins were subjected to multiple sequence alignment using DNAMAN, and the characteristic structural domains, namely APB, APA, and bHLH, were marked. Different colors are used to indicate the similarity of multiple sequence alignments. (Black = 100%, pink > 75%, blue > 50%, yellow > 30%) (**C**) Motif analysis on PIF proteins used MEME, and the 12 analyzed motifs are visualized on the right. Chiclop is used to beautify the images. The protein sequences used for analysis are TaPIF1-1A (TraesCS1A02G083000.1), TaPIF1-1B (TraesCS1B02G100400.1), TaPIF1-1D (TraesCS1D02G084200.1), TaPIF2-1A (TraesCS1A02G212700.1), TaPIF2-1B (TraesCS1B02G226200.1), TaPIF2-1D (TraesCS1D02G215600.1), TaPIF3-2A (TraesCS2A02G253900.1), TaPIF3-2B (TraesCS2B02G273500.1), TaPIF3-2D (TraesCS2D02G254400.1), TaPIF4-5A (TraesCS5A02G049600.1), TaPIF4-5B (TraesCS5B02G054800.1), TaPIF4-5C (TraesCS5D02G060300.1), TaPIF5-5A (TraesCS5A02G376500.1), TaPIF5-5B (TraesCS5B02G380200.1), TaPIF5-5D (TraesCS5D02G386500.1), TaPIF6-5A (TraesCS5A02G420200.1), TaPIF6-5B (TraesCS5B02G422000.1), PtPIF1 (Potri.002G252800.9), PtPIF3a (Potri.005G001800.1), PtPIF3b (Potri.013G001300.4), PtPIF4/5a (Potri.002G055400.11), PtPIF4/5b (Potri.005G207200.12), PtPIF8a (Potri.002G143300.1), PtPIF8b (Potri.014G066500.1), PtPIF9a (Potri.005G139700.2), PtPIF9b (Potri.014G025800.1), PtPIF10 (Potri.014G111400.1), CmPIF1 (MELO3C012808.2), CmPIF3 (MELO3C031303.2), CmPIF4 (MELO3C026410.2), CmPIF8 (MELO3C022233.2), GhPIF1-1a (Gh_A11G1248.1), GhPIF1-1d (Gh_D11G1395), GhPIF1-2d (Gh_D07G1543), GhPIF1-3d (Gh_D05G3213), GhPIF1-4d (Gh_D07G2050.1), GhPIF2a (Gh_A11G3067), GhPIF2d (Gh_D11G1107), GhPIF3a (Gh_A11G2494), GhPIF3d (Gh_D11G2839), GhPIF6a (Gh_A07G1202), GhPIF6d (Gh_D07G1303), GhPIF7-1 (Gh_A03G0607), GhPIF7-1d (Gh_D03G0895), GhPIF7-2d (Gh_D07G0698), GhPIF8a (Gh_A11G0710), GhPIF8d (Gh_D11G0826), GhPIF9-1a (Gh_A07G0148), GhPIF9-1d (Gh_D07G0141), GhPIF9-2d (Gh_D09G2368), GhPIF9-3d (Gh_D11G1337), GhPIF9-4a (Gh_A08G1091), GhPIF9-4d (Gh_D08G1374), GhPIF9-5a (Gh_A10G1164), GhPIF9-5d (Gh_D10G1331), AtPIF1 (AT2G20180.2), AtPIF3 (AT1G09530.1), AtPIF4 (AT2G43010.5), AtPIF5 (AT3G59060.2), AtPIF6 (AT3G62090.2), AtPIF7 (AT5G61270.1), OsPIL11 (Os12g0610200), OsPIL12 (Os03g0639300), OsPIL13 (Os03g0782500), OsPIL14 (Os07g0143200), OsPIL15 (Os01g0286100), OsPIL16 (Os05g0139100), ZmPIF1 (GRMZM2G115960_P03), ZmPIF3 (Zm00001eb332400_P001), ZmPIF5.2 (Zm00001eb213550_P001), ZmPIF5.1 (Zm00001eb059460_P001), ZmPIF4.2/5 (Zm00001eb050790_P001), ZmPIF4/4.1 (Zm00001eb031560_P001), ZmPIF3.3 (Zm00001eb417610_P001), SlPIF1a (XP_004247109.1), SlPIF1b (XP_004240467.1), SlPIF3 (XP_010313958.1), SlPIF4/5 (XP_004243631.1), SlPIF7 (XP_004242180.1), SlPIF8 (XP_004229781.1), MdPIF1 (MD10G1170600), MdPIF2 (MD04G1185100), MdPIF3 (LOC103450807), MdPIF4 (MD17G1132600), MdPIF5 (MD09G1146000), MdPIF7 (MD14G1208000), MdPIF8 (MD07G1113200).

**Figure 2 ijms-25-02198-f002:**
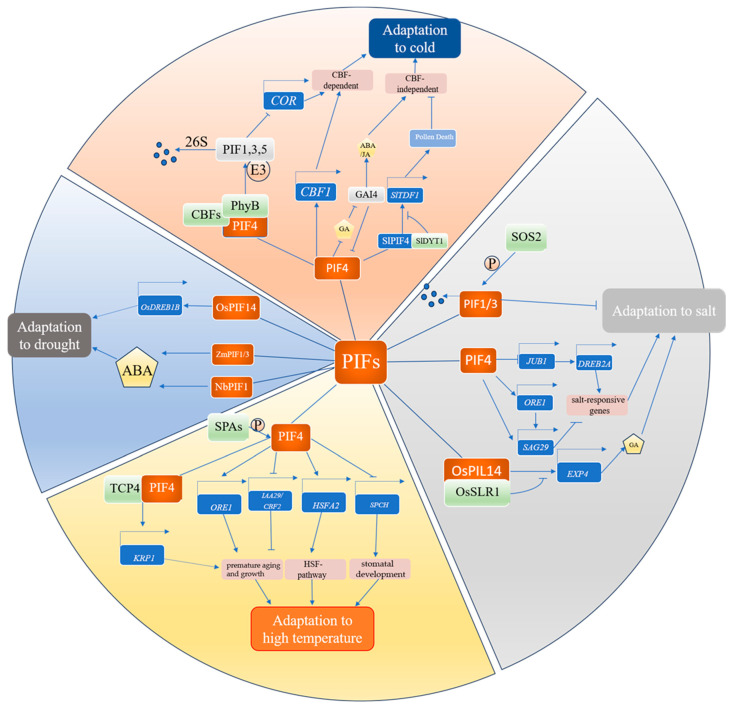
A typical molecular model demonstrates how PIFs regulate adaptability to abiotic stress. PIFs, responsive to abiotic stress, can be induced to express or undergo phosphorylation by other protein kinases, thus perceiving abiotic stress signals. Following this, PIFs convey these signals downstream, initiating a multitude of signaling cascades. A symphony of secondary signals and plant hormones collaboratively fine-tune a sophisticated molecular mechanism, triggering a comprehensive array of regulatory elements. This orchestrates the holistic transcriptional reprogramming of a diverse range of genes associated with abiotic stress. In the model diagram, red rounded rectangles represent different PIF proteins, light green rounded rectangles represent different PIF interacting proteins, and the blue rounded rectangles indicate target genes directly regulated by PIFs. Symbols: ˧ indicates negative regulation; → indicates positive regulation. Abbreviations: *OsDREB1B*, *oryza sativa dehydration responsive element binding*; *COR*, *cold-responsive gene*; *CBF*, *C-repeat binding factors*; GAI4, gibberellic acid insensitive 4; *SlTDF*, *tapetal development and function 1*; SlDYT, dysfunctional tapetum 1; SPAs, suppressor of phyA-105; TCP4, teosinte branched1/cycloidea/pcf4; *KPR1*, *KIP-related protein 1*; *HSFA2*, *heat shock factor A2*; *ORE1, oresara1*; *IAA29*, *indole-3-acetic acid inducible 29*; *SPCH*, *SPEECHLESS*; SOS2, salt overly sensitive 2; *JUB1*, *jungbrunnen 1*; *SAG29*, *senescence-associated gene 29*; *OsSLR1*, *slender rice 1*; *EXP4*, *expansin 4*.

**Figure 3 ijms-25-02198-f003:**
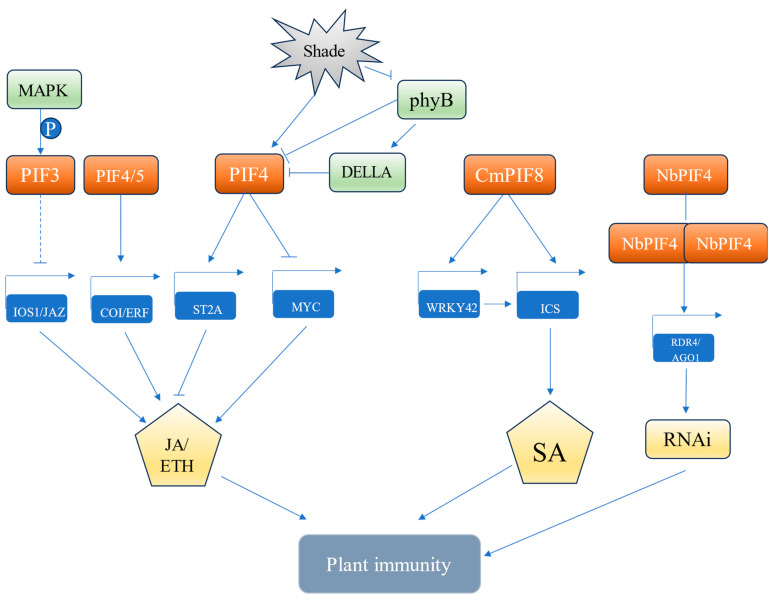
A schematic model illustrates how PIFs counteract biotic stress through various pathways. PIFs play a crucial role in plant immunity by directly regulating downstream defense-related genes and participating in the signal transduction of classic defense pathways, such as JA/ETH and SA, thereby orchestrating the plant’s immune response. In the model diagram, red rounded rectangles represent different PIF proteins; light green rounded rectangles represent different PIF interacting proteins; and blue rounded rectangles indicate target genes directly regulated by PIF binding. Symbols: ˧ indicates negative regulation; → indicates positive regulation. Abbreviations: MAPK, mitogen-activated protein kinases; *IOS1*, *iron-deficiency overly sensitive*; *JAZ*, *jasmonate ZIM-domain*; *COI*, *coronatine insensitive 1*; *ERF*, *ethylene responsive factor*; *ST2A*, *sulfotransferase*; MYC, myelocytomatosis; ICS, isochorismate synthate; RDR6, *RNA-dependent RNA polymerase 6*; *AGO1*, *argonaute 1*.

**Table 1 ijms-25-02198-t001:** PIF transcription factors have been reported to participate in various biotic and abiotic stress responses.

Stress Conditions	PIF TFs	Associated Responses	References
Drought	OsPIF14, OsPIL1,	DREB-mediated pathway	[40,41,42]
	ZmPIF1, ZmPIF3, DcPIF3, MfPIF1, NbPIF1/NtPIF1	ABA signaling pathways	[43,44,45,46,47,48]
Low temperature	SlPIF4, OsPIL16, AtPIF1, AtPIF3, AtPIF4, AtPIF5, AtPIF7	CBF-dependent regulatory pathways	[49,50,51,52,53]
	SlPIF4, CsPIF8, MdPIF3	CBF-independent regulatory pathways	[49,54,55,56]
High temperature	AtPIF4	HSF-mediated heat stress response pathway	[57]
	AtPIF4, AtPIF5	Premature aging and growth caused by high temperature	[58,59,60,61,62]
	AtPIF4	Stomatal development	[63,64,65]
Salt	AtPIF4, OsPIL14	Transcriptional regulation of salt-responsive genes	[66,67,68,69]
	AtPIF1/3, AtPIF4/5	SOS pathway	[70,71]
	MfPIF1/8, CaPIF8	GA signaling pathways	[47,72,73]
Biotic stress	AtPIF3, AtPIF4, AtPIF5	JA/ETH	[74,75,76,77]
	CmPIF8	SA	[78]
	NbPIF4	RNAi	[79]

## Data Availability

Not applicable.

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
