# Peer review of "Functions of Phytochrome Interacting Factors (PIFs) in Adapting Plants to Biotic and Abiotic Stresses"

_ijms, 2024, doi:10.3390/ijms25042198_

Round 1
Reviewer 1 Report
Comments and Suggestions for Authors
The review by Li et al. summarizes the functions of PIFs in various stresses of plants. The review provides a good breakdown of the multifaceted functions of PIFs in plants, however, certain specific comments are addressed below and should be revised.
1. A table can be added in this manuscript to list the roles and references of PIFs in various stresses of plants.
2. The resolution of the figure is relatively low and needs to be improved.
3. Further description is needed for the structures of PIFs from different plants, can also be pointed in a figure.
4. The phylogenetic relationship analysis in figure 1 should be re-performed, using more PIFs from different plants, and the names of groups should be added.
5. A summarily and forward-looking figure about PIFs (including the interacted proteins and regulating genes) is needed in this manuscript.
6. Please check the manuscript for typo errors, grammatical and spelling mistakes carefully. For example, line 4: "†" should be deleted; Line 24: "abiotics" should be “abiotic”. Line 44: "anActive". Line 194: CBF2. Line 201: “Speechles” should be “SPEECHLESS”. Some other mistakes were not present here. It is suggested to revise the manuscript and correct the technical and grammatical mistakes.
Comments on the Quality of English LanguageIt is suggested to revise the manuscript and correct the technical and grammatical mistakes.
Author Response
Response to Reviewer1
We are grateful of your valuable suggestions for improving the quality of our manuscript. We have addressed all of your comments and please see the point-to-point response in the following section. A revised manuscript has been updated in which the revisions or corrections made are highlighted or tracked.
Comments 1: A table can be added in this manuscript to list the roles and references of PIFs in various stresses of plants.
We appreciate the reviewer’s suggestion. We have organized a table in which PIFs were classified into groups based on their functions. The related regulating pathways and the references were also included. Please see the new Table 1 for detail.
Comments 2: The resolution of the figure is relatively low and needs to be improved.
Thank you for your comment. We have improved the quality of the figures.
Comments 3: Further description is needed for the structures of PIFs from different plants, can also be pointed in a figure.
We appreciate your suggestions and we agree with your suggestions. We have improved the description of the PIFs' structure, please see lines 49-60 in our revised version. We have also added a schematic illustration of the motif structure of PIFs, please see the revised Figure 1 for detail.
Comments 4: The phylogenetic relationship analysis in figure 1 should be re-performed, using more PIFs from different plants, and the names of groups should be added
Thank you for your suggestion. We have collected more PIFs from plants and a new phylogenetic tree analysis using PIFs from a broader range of species was constructed, andthese PIFs can be categorized into groups (Group I-V), as displayed in revised Figure 1.
Comments 5: A summarily and forward-looking figure about PIFs (including the interacted proteins and regulating genes) is needed in this manuscript.
Thank you for your insightful suggestion. We have incorporated the forward-looking summary on PIFs into lines 413-425of the document, as you advised.
Comments 6: Please check the manuscript for typo errors, grammatical and spelling mistakes carefully. For example, line 4: "†" should be deleted; Line 24: "abiotics" should be “abiotic”. Line 44: "anActive". Line 194: CBF2. Line 201: “Speechles” should be “SPEECHLESS”. Some other mistakes were not present here. It is suggested to revise the manuscript and correct the technical and grammatical mistakes.
Thank you for pointing out the errors. We have extensively polished the language of our manuscript, and revised the mentioned errors in the new version.

Reviewer 2 Report
Comments and Suggestions for Authors
The review on "Functions of phytochrome interacting factors (PIFs) in adapting plants to biotic and abiotic stresses” is intriguing, but there are a few queries that need to be addressed:
1. Authors mentioned OsPIL without abbreviation; please provide its full form.
2. In line number 103, "Therefore, PIFs are among the key regulators that adapting plants to arid environments by regulating various signaling pathways, and DREB and ABA signaling pathways are among them," authors need to clarify if DREB is a signaling pathway.
3. Please correct it as “Therefore, PIFs are among the key regulators that adapt plants to arid environments by regulating various signaling pathways, like ABA."
4. Please elucidate how many PIFs are there in plants, and which ones are crucial?
5. When using an abbreviation for the first time, mention its full form in brackets. For example, in line 116, SlPIF4.
6. In Figure 2, authors mentioned PIF interacting genes and their adaptation to stresses. Does it occur without signaling pathways? Genes alone may not influence total stress; they should mediate through some signaling pathways. Please clarify in Figure 2. Except for drought, there are no signaling pathways, indicating vague information.
7. In Figure 3, authors included RNAi. Is it a signaling pathway? What is NbPIF4? Its role is not mentioned anywhere in the text; please provide a clear explanation.

there are grammatical mistakes, mentioned in comments, one of the example.
Author Response
Response to Reviewer2
We are grateful of your valuable suggestions for improving the quality of our manuscript. We have addressed all of your comments and please see the point-to-point response in the following section. A revised manuscript has been updated in which the revisions or corrections made are highlighted or tracked.
Comments 1: Authors mentioned OsPIL without abbreviation; please provide its full form.
We thank the reviewer’s suggestion. OsPIL1 has been corrected to OsPIL1 (Oryza sativa phytochrome-interacting factor-like 1).
Comments 2: In line number 103, "Therefore, PIFs are among the key regulators that adapting plants to arid environments by regulating various signaling pathways, and DREB and ABA signaling pathways are among them," authors need to clarify if DREB is a signaling pathway.
We thank the reviewer for picking this out. We have revised this sentence as suggested by the reviewer. Line 157-158
Comments 3: Please correct it as “Therefore, PIFs are among the key regulators that adapt plants to arid environments by regulating various signaling pathways, like ABA.
We thank the reviewer for picking this out. We have revised this sentence as suggested by the reviewer. Line 157-158
Comments 4:Please elucidate how many PIFs are there in plants, and which ones are crucial?
We appreciate the reviewer’s comment. According to the published reports up to now, PIFs have been identified in many plant species, including Arabidopsis (6 PIFs), hexaploid wheat (18 PIFs), rice (6 PIFs), in maize (7 PIFs), tomato (8 PIFs), apple (7 PIFs), poplar (10 PIFs), melon (4 PIFs), and cotton (24 PIFs). Moreover, with the improvement of sequencing technology, more PIFs will be identified in many other plant species, especially in crop plants.
For the second question, PIF4 is the most extensively studied member with a wide range of functions in multiple plants. In addition to function as key component of light-signaling transduction, it also plays a role in regulating plant development of various organs such as pollen, and closely associates with regulation of low temperature, high temperature, and salt stress and biotic stress responses. Of course, other PIFs, such PIF1 and PIF3 are also key factor involved in light-signaling transduction.
Comments 5:When using an abbreviation for the first time, mention its full form in brackets. For example, in line 116, SlPIF4.
We thank the reviewer’s comment. SlPIF4 has been corrected to SlPIF4 (Solanum lycopersicum phytochrome interacting factor 4), and similar issued have been revised all through the manuscript.
Comments 6:In Figure 2, authors mentioned PIF interacting genes and their adaptation to stresses. Does it occur without signaling pathways? Genes alone may not influence total stress; they should mediate through some signaling pathways. Please clarify in Figure 2. Except for drought, there are no signaling pathways, indicating vague information.
We appreciate the reviewer’s suggestion. We can not agree more that genes will function through signaling pathways. Similar to drought, we have included signaling pathways in adaption to other stresses, please see the revised figure 2. Moreover, we also generated table 1, in which the functions of PIFs in different stresses, as well as the related response pathways were presented.
Comments 7:In Figure 3, authors included RNAi. Is it a signaling pathway? What is NbPIF4? Its role is not mentioned anywhere in the text; please provide a clear explanation.
We thank the reviewer’s question. RNA interference (RNAi) is a defense mechanism that plays a critical role in against viruses in in plants. NbPIF4 indicates the PIF4 ortholog in Nicotiana benthamiana. Here, we considered NbPIF4 mediated RNAi as a response pathway that mediating the interactions between plants and viruses. We have added the related information regarding the involvement of NbPIF4 in regulating RNAi in lines 381-386 of our revised manuscript.

Reviewer 3 Report
Comments and Suggestions for Authors
The manuscript is prepared for a current problem (topic), but before its publication it is advisable to supplement the text with some current information and minor formal adjustments to the text.
On the formal side, I recommend careful proofreading of the terms used. e.g. Latin (botanical) names with italics, somewhere proteins and genes are interchanged, i.e. genes are written in italics, proteins are not written in italics - this distinction is useful for understanding the content of the text and the focus of the review. Line 128 Arabidopsis Thaliana - is correctly Arabidopsis thaliana. Line 331 summary is correctly Summary, etc.
In terms of content, the text is logically divided into individual chapters, but current publications are not always cited here. Therefore, I take the liberty of suggesting, for example, the insertion of three articles published in 2023, which are related to the given problem and appropriately connect findings. In chapter 3. PIF and CBF, it would be appropriate to insert the publication and information that CBF play a significant role in resistance to abiotic stress in grapevine (Vitis) /Fang et al., 2023 - DOI: 10.17221/82/2022-CJGPB/. In chapter 6, the authors describe PIF and biotic stress, e.g. with MAPK it is not only biotic stress, but also abiotic stress together with other factors /bLHL, bZIP/, e.g. in barley - Alamholo and Tarinejad, 2023 - DOI: 10.17221/ 69/2022-CJGPB; or cytochrome P450 /CYP/ and interaction with ABA affects drought tolerance in cotton /Gu et al., 2023 - DOI: 10.17221/108/22022-CJGPB/. At the same time, examples of important representatives of agricultural crops will deepen the importance of the study and its necessity. That is impacts not only of findings in Arabidopsis thaliana. It is also important to note that some PIF-interacting regions themselves individually influence resistance to biotic and abiotic stress. Possible interaction with PIF can increase the importance of these regions of the genome and thus be an area of interest not only for theoretical studies in model organisms, but mainly for practical applications (plant breeding, etc.).
In the case of figures that are part of the manuscript, they would deserve a much better legend, e.g. Figure 1 is divided into part (a) and (b), but this is not taken into account in the title and legend of this figure. Similarly for Figure 2 and 3 - there is a lack of a self-explanatory legend that would explain all the abbreviations used.
In the references section, it would be good to check all the names of the journal, whether their writing of the names corresponds to the standards, e.g. line 665, 687, 702, 748, etc.
Based on the above, I recommend the manuscript for publication after major revision and second review.
Comments on the Quality of English Language
At the same time, I recommend a careful check of the English language to eliminate minor typos and not exactly appropriate wording.
Author Response
Response to Reviewer3
We are grateful of your valuable suggestions for improving the quality of our manuscript. We have addressed all of your comments and please see the point-to-point response in the following section. A revised manuscript has been updated in which the revisions or corrections made are highlighted or tracked.
Comments 1: On the formal side, I recommend careful proofreading of the terms used. e.g. Latin (botanical) names with italics, somewhere proteins and genes are interchanged, i.e. genes are written in italics, proteins are not written in italics - this distinction is useful for understanding the content of the text and the focus of the review. Line 128 Arabidopsis Thaliana - is correctly Arabidopsis thaliana. Line 331 summary is correctly Summary, etc.
We thank the reviewer’s suggestion. We have corrected all the related issues mentioned by the reviewer, and we have extensively polished the language of our manuscript.
Comments 2: In terms of content, the text is logically divided into individual chapters, but current publications are not always cited here. Therefore, I take the liberty of suggesting, for example, the insertion of three articles published in 2023, which are related to the given problem and appropriately connect findings. In chapter 3. PIF and CBF, it would be appropriate to insert the publication and information that CBF play a significant role in resistance to abiotic stress in grapevine (Vitis) /Fang et al., 2023 - DOI: 10.17221/82/2022-CJGPB/. In chapter 6, the authors describe PIF and biotic stress, e.g. with MAPK it is not only biotic stress, but also abiotic stress together with other factors /bLHL, bZIP/, e.g. in barley - Alamholo and Tarinejad, 2023 - DOI: 10.17221/ 69/2022-CJGPB; or cytochrome P450 /CYP/ and interaction with ABA affects drought tolerance in cotton /Gu et al., 2023 - DOI: 10.17221/108/22022-CJGPB/.
We appreciated the reviewer’s suggestion. All these newly published literature have been incorporated in our revised manuscript.
At the same time, examples of important representatives of agricultural crops will deepen the importance of the study and its necessity. That is impacts not only of findings in Arabidopsis thaliana. It is also important to note that some PIF-interacting regions themselves individually influence resistance to biotic and abiotic stress. Possible interaction with PIF can increase the importance of these regions of the genome and thus be an area of interest not only for theoretical studies in model organisms, but mainly for practical applications (plant breeding, etc.).
We can not agree more that presenting representatives of agricultural crops will deepen the importance of the study. In this manuscript we have mentioned the function of PIFs in rice (line 124-128; line134-139; line 306-312), tomato (line 186-198), apple (line 204-205), pepper (line 303-305), oriental melon (line 342-345). We believe that more PIFs will be identified to be critical for adaption to various stresses of agricultural crops.
Comments 3: In the case of figures that are part of the manuscript, they would deserve a much better legend, e.g. Figure 1 is divided into part (a) and (b), but this is not taken into account in the title and legend of this figure. Similarly for Figure 2 and 3 - there is a lack of a self-explanatory legend that would explain all the abbreviations used.
We appreciate the reviewer’s comment. We have supplemented the captions of the figures in our new version.
Comments 4: In the references section, it would be good to check all the names of the journal, whether their writing of the names corresponds to the standards, e.g. line 665, 687, 702, 748, etc.
We thank the reviewer for picking these out. We have checked all of our references and revised the related issues.

Round 2
Reviewer 1 Report
Comments and Suggestions for Authors
Thanks for the author's serious response. This manuscript meets the requirements of this journal and can be published.
Author Response
Thank you for your efforts in improving the quality of our manuscript.
Reviewer 2 Report
Comments and Suggestions for Authors
All the comments are addressed. Satisfied with authors comments
Comments on the Quality of English Languageminor English corrections required
Author Response
感谢您为提高我们稿件的质量所做的努力。
Reviewer 3 Report
Comments and Suggestions for Authors
The authors accepted most of my comments. I really recommend a careful formal review of the text, because there are still little things that need to be edited. Below are a few examples:
line 117 - Abscisic Acid - correctly abscisic acid;
line 147 - Daucus carota L. - "L." is not written in italics;
line 176 - bzr-1D...is part of the gene name, i.e. properly it should be italics;
line 178, 293, 296, etc. - Arabidopsis - correctly in italics;
line 186 - Gibberellin (GA) - properly gibberellin.
That's why I recommend to the authors a careful formal proofreading. More of these minor mistakes and inaccuracies lower the level of an otherwise high-quality manuscript. Based on the above, I recommend accepting the manuscript after minor revision.
Comments on the Quality of English LanguageThe manuscript needs careful checking due to the correction of upper and lower case letters and standards when using italics.
Author Response
The authors accepted most of my comments. I really recommend a careful formal review of the text, because there are still little things that need to be edited. Below are a few examples:
line 117 - Abscisic Acid - correctly abscisic acid;
line 147 - Daucus carota L. - "L." is not written in italics;
line 176 - bzr-1D...is part of the gene name, i.e. properly it should be italics;
line 178, 293, 296, etc. - Arabidopsis - correctly in italics;
line 186 - Gibberellin (GA) - properly gibberellin.
That's why I recommend to the authors a careful formal proofreading. More of these minor mistakes and inaccuracies lower the level of an otherwise high-quality manuscript. Based on the above, I recommend accepting the manuscript after minor revision.
Thank you for your careful review of the manuscript. I deeply regret some minor errors in the text. I have carefully corrected the errors you mentioned, marked in red to ensure clarity. In addition, I carefully verified that the capitalization and italics of the text were correct.
